# PROMPT-TUNING DECISION TRANSFORMER WITH PREFERENCE RANKING

## ABSTRACT

Prompt-tuning has emerged as a promising method for adapting pre-trained models to downstream tasks or aligning with human preferences. Prompt learning is widely used in NLP but has limited applicability to RL due to the complex physical meaning and environment-specific information contained within RL prompts. Directly extending prompt-tuning approaches to RL is challenging because RL prompts guide agent behavior based on environmental modeling and analysis, rather than adjusting the prompt format for downstream tasks widely used in NLP. In this work, we propose the Prompt-Tuning DT algorithm to address these challenges by using trajectory segments as prompts to guide RL agents in acquiring environmental information and optimizing prompts via black-box tuning to enhance their ability to contain more relevant information, thereby enabling agents to make better decisions. Our approach involves randomly sampling a Gaussian distribution to fine-tune the elements of the prompt trajectory and using the preference ranking function to find the optimization direction, thereby providing more informative prompts and guiding the agent toward specific preferences in the target environment. Extensive experiments show that with only 0.03% of the parameters learned, Prompt-Tuning DT achieves comparable or even better performance than full-model fine-tuning in the few-shot settings. Our work contributes to the advancement of prompt-tuning approaches in RL, providing a promising direction for optimizing pre-trained large-scale RL agents for specific preference tasks.

## 1 INTRODUCTION

Pre-trained large-scale models (PLMs) (Brown et al., 2020; Devlin et al., 2018; Lee et al., 2022; Reed et al., 2022) have proven to be highly effective for a wide range of tasks due to their high transferability and competitive performance on downstream tasks with limited target data. However, full-model fine-tuning requires updating and storing all the parameters of the PLM, which is memory-intensive and impractical for maintaining a separate set of parameters for each task during inference. Recently, prompt-tuning (Li & Liang, 2021; Shin et al., 2020) has emerged as a promising alternative to full-model fine-tuning, allowing for the effective adaptation of pre-trained models to specific downstream tasks and human preferences. By freezing the pre-trained model's parameters and tuning only the prompts, prompt-tuning approaches have demonstrated comparable performance with full-model fine-tuning methods across various model scales and tasks (Liu et al., 2021; Lester et al., 2021; Zhong et al., 2021).

Offline Reinforcement Learning (offline RL) is a data-driven approach that learns an optimal policy from trajectories collected by a set of behavior policies, without requiring access to the environments. This approach is critical in many settings, where online interactions are expensive or dangerous. However, offline RL struggles with generalization to unseen tasks and adapting to preferences, as the agent may not find a good policy in the test tasks due to the distribution shift. Recent works address this challenge through offline meta-RL, which leverages the algorithmic learning perspective (Mitchell et al., 2021; Nichol et al., 2018; Rajeswaran et al., 2019). In contrast, we aim to investigate the power of prompt-tuning methods with PLMs. Nonetheless, unlike natural language processing (NLP) prompts, RL prompts are more complex and contain environment-specific information, which may be vulnerable to the prompt learning process. Additionally, prompt-tuning approaches from NLP cannot be directly applied to RL prompts, as RL prompts guide agent behavior based on environmental modeling and analysis rather than adjusting the prompt format for downstream tasks.

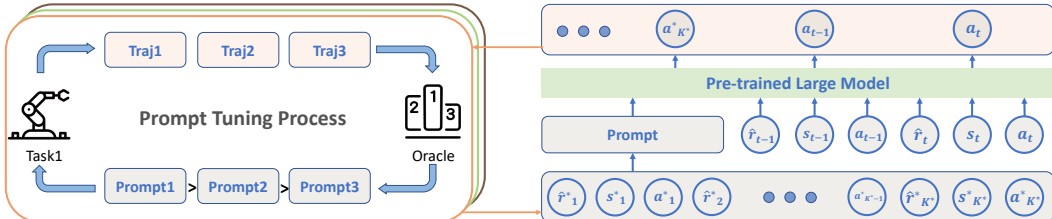

Figure 1: Application of Prompt-Tuning DT. At each iteration, the Pre-trained Large Model generates different trajectories for the current task based on different prompts, which are generated by perturbing the initial prompt with random Gaussian distribution, and the most recent K-step history. The generated trajectories are then ranked based on a specific property using a preference ranking oracle, and the ranking information is leveraged to update the prompt.

Therefore, there is an urgent need to develop novel prompt-tuning techniques specifically tailored to RL that can guide agents toward specific preferences in the target environment.

In this paper, we propose a novel algorithm called Prompt-Tuning Decision Transformer (Prompt-Tuning DT) as an approach to tackle the challenge of generalization in offline RL from the perspective of prompt tuning. Our approach leverages trajectory segments as prompts to guide RL agents in acquiring target environmental information and optimizes prompts via black-box tuning to enhance their ability to contain more meaningful information. Prompt-tuning is essential in RL as it enables pre-trained agents to make better decisions by providing more informative prompts. This contrasts with the limitations inherent in straightforward prompt-based adaptation methods (Xu et al., 2022): The process of generating high-quality trajectory prompts involves significant investments of time and resources, and the prompt's effectiveness is constrained by the model's input capacity for conditioning prompts (Lester et al., 2021). As a result, despite the progress made in prompt-based adaptation, downstream task quality still lags far behind that of full-model fine-tuned methods.

In our prompt-tuning offline RL framework, we first pre-train the agent using offline trajectories from various tasks within the same environment. For each task, the agent learns to predict a target trajectory by conditioning on a trajectory prompt sampled from the same task. During the evaluation, the agent is presented with a new task and a small set of new trajectories for fine-tuning the prompt (total step length not exceeding 10). Our approach perturbs each element of the prompt by randomly sampling from a Gaussian distribution to avoid catastrophic deviations and employs a preference ranking function along with a ranking algorithm to determine the optimization direction. Notably, by optimizing a mere 0.03% of the model parameters, Prompt-Tuning DT achieves performance on par with full-model fine-tuning and surpasses alternative parameter-efficient methods. Our work contributes to the advancement of prompt-tuning approaches in RL, providing a promising direction for optimizing PLMs for specific preferences and downstream tasks.

In summary, our main contributions are three-fold:

- We propose Prompt-Tuning DT, a memory-efficient alternative to fine-tuning pre-trained agents that achieves comparable performance with full-model fine-tuning methods.

- We present a prompt-tuning RL framework, which leverages a PLM's API to enable streamlined customization for specific preferences with minimal parameter modifications.

- By optimizing with preference ranking, Prompt-Tuning DT outperforms strong meta offline RL baselines, demonstrating its effectiveness as a few-shot learner for generalization in offline RL.

## 2 RELATED WORK

**Offline RL.** Offline RL has emerged as a promising paradigm for learning from fixed, limited datasets consisting of trajectory rollouts from arbitrary policies (Levine et al., 2020). However, deploying off-policy RL algorithms directly in the offline setting can be challenging due to the distributional shift problem, which can result in a significant performance drop (Fujimoto et al., 2019). To overcome this issue, model-free RL algorithms adopt various strategies, such as constraining the action space of the policy (Kumar et al., 2019) or introducing pessimism to the value function

(Kumar et al., 2020), to explicitly correct the distributional mismatch between the behavior policy in the offline data and the optimized policy. In contrast, model-based RL algorithms estimate the environmental reward and transition functions using offline data and require modifications to the RL algorithm to avoid exploiting errors in the estimated model (Kidambi et al., 2020; Yu et al., 2020b).

Offline RL has been increasingly viewed as a sequence modeling task, and Transformer-based decision models have been applied to this domain. The objective is to predict the next sequence of actions based on the sequence of recent experiences, which includes state-action-reward triplets. This approach can be trained using supervised learning, which makes it more suitable for offline RL and imitation learning. Several studies have explored the use of Transformers in RL under the sequence modeling paradigm, including Gato (Reed et al., 2022), Generalized DT (Furuta et al., 2021), Graph DT (Hu et al., 2023), and the survey works (Hu et al., 2022; Li et al., 2023). In this work, we propose a novel approach that is based on Prompt-DT (Xu et al., 2022) and incorporates prompt-tuning techniques to enhance its performance on downstream target tasks.

**Meta RL.** Meta-learning algorithms enable efficient learning of new tasks by learning the process of learning itself. In the context of meta-RL, the objective is to generalize an agent's knowledge from one task to another. In recent years, several studies have delved into the problem of offline meta-RL. Li et al. (2020) address a scenario where task identity is spuriously inferred due to biased datasets and apply the triplet loss to robustify the task inference with reward relabelling. Dorfman et al. (2021) extend the online Meta-RL method VariBAD (Zintgraf et al., 2019) to the offline setup, where they assume known reward functions for each task and use reward relabelling to share data across tasks with shared dynamics. On the other hand, Mitchell et al. (2021) propose an offline Meta-RL algorithm based on MAML (Finn et al., 2017). Their approach includes an advantage weighting loss and introduces a policy update in the inner loop to theoretically increase the richness of the policy's update and empirically improve adaptation performance and stability. In this article, we investigate an alternative perspective on meta-RL using sequence modeling and prompt engineering, which can achieve comparable or superior performance compared to traditional methods.

**Prompt Learning.** Prompt learning is a promising methodology in NLP that involves optimizing a small subset of parameters while leaving the main model architecture unchanged. The basic premise of prompt learning involves presenting the model with a cloze test-style textual prompt, which the model is then expected to fill in with the corresponding answer. Autoprompt (Shin et al., 2020) proposes an automatic prompt search methodology for efficiently finding optimal prompts, while recent advancements in prompt learning have adopted trainable continuous embeddings for prompt representation (Li & Liang, 2021; Lester et al., 2021). Prompt learning has also been applied to the vision-language domain, where introducing continuous prompts into pre-trained vision-language models has led to significant improvements in few-shot visual recognition and generalization performance (Zhou et al., 2022b;a). While prompt learning reduces the number of tunable parameters, back-propagation through the entire model is still necessary to calculate gradients and update the small subset of parameters. Gradient-free methods have been proposed to optimize continuous (Sun et al., 2022) or discrete (Prasad et al., 2022) prompts. Despite the great success of prompt-tuning in the fields of NLP and CV, its application in RL has not been thoroughly explored. Therefore, in this study, we propose the Promp-Tuning DT method that employs gradient-free methods to optimize continuous trajectory prompts with a preference ranking oracle. This approach can be extended to a human-in-the-loop environment, where candidate prompts are ranked manually.

## 3 PRELIMINARY

In this section, we provide a concise overview of the key components of our algorithm, namely decision transformer and ranking optimization. Decision transformer extends the Transformer model to offline RL by framing RL problems as sequence modeling tasks, paving the way for the potential emergence of large-scale RL models. We also introduce a ranking optimization approach that utilizes ranking data to optimize the model without explicitly calculating gradient information. These algorithms form the basis of our approach illustrated in Section 4.

### 3.1 DECISION TRANSFORMER

Transformer (Vaswani et al., 2017), extensively studied in NLP (Devlin et al., 2018) and CV (Dosovitskiy et al., 2020), has also been explored in RL using the sequence modeling pattern (Hu et al.,

2022). Moreover, Recent works from NLP suggest Transformers pre-trained on large-scale datasets are capable of few-shot or zero-shot learning under the prompt-based framework (Liu et al., 2023; Brown et al., 2020). Building upon this, Gato (Reed et al., 2022) and TTP (Jain et al., 2023) both extend the prompt-based framework to the offline RL setting, constructing pre-trained large-scale agents designed to address multiple tasks concurrently in a zero-shot or few-shot fashion. Both methods are based on the Decision Transformer (Chen et al., 2021) which treats learning a policy as a sequence modeling problem. DT introduces the notion of modeling trajectories through state $s_t$, action $a_t$, and reward-to-go $\hat{r}_t$ tuples collected at distinct time steps $t$. The reward-to-go token quantifies the cumulative reward from the current time step to the end of the episode. During training with offline collected data, DT processes a trajectory sequence $\tau_t$ in an auto-regressive manner which encompasses the most recent K-step historical context.

$$\tau_t = (\hat{r}_{t-K+1}, s_{t-K+1}, a_{t-K+1}, \ldots, \hat{r}_t, s_t, a_t). \tag{1}$$

The prediction head associated with a state token $s_t$ is trained to predict the corresponding action $a_t$. Regarding continuous action spaces, the training objective is to minimize the mean-squared loss:

$$L_{DT} = \mathbb{E}_{\tau_t \sim \mathcal{T}} \left[ \frac{1}{K} \sum_{i=t-K+1}^{t} (a_i - \pi(\tau_t)_i)^2 \right]. \tag{2}$$

## 3.2 RANKING OPTIMIZATION

Black-box optimization, which utilizes a derivative-free framework to optimize the target function, has been extensively studied in the optimization literature for several decades (Frazier, 2018; Nesterov & Spokoiny, 2017; Loshchilov & Hutter, 2016). With the rapid development of Reinforcement Learning with Human Feedback (RLHF), ranking data, which enables humans to express their personal preferences in a straightforward and intuitive manner (Bai et al., 2022; Ouyang et al., 2022), has demonstrated great potential for use in various applications, especially those where the exact value of personal information is sensitive, such as healthcare or finance. ZO-RankSGD (Cai et al., 2022; Bergou et al., 2020; Tang et al., 2023) is an effective approach for model optimization that finds the descent direction directly from the ranking information, without the need for knowledge of the gradient of the model or the exact value of the data. Given a ranking function $f : \mathbb{R}^d \to \mathbb{R}$ and $m$ points $x^1, \ldots, x^m$ to query, an $(m, k)$ ranking oracle $O_f^{(m,k)}$ will return $k$ smallest points sorted in their order. With the ranking oracle $O_f^{(m,k)}$ and a starting point $x$, we can query $O_f^{(m,k)}$ with the inputs $x^i = x + \mu \xi_i$, $\xi_i \sim \mathcal{N}(0, I_d)$, for $i = 1, \ldots, m$, and $\mu$ is a constant. With the directed acyclic graph (DAG) $\mathcal{G} = (\mathcal{N}, \mathcal{E})$ constructed from the ranking information of $O_f^{(m,k)}$, where the node set $\mathcal{N} = \{1, \ldots, m\}$ and the directed edge set $\mathcal{E} = \{(i, j) \mid f(x_i) < f(x_j)\}$, the rank-based gradient estimator can be formulated as follows:

$$\tilde{g}(x) = \frac{1}{|\mathcal{E}|} \sum_{(i,j) \in \mathcal{E}} \frac{x^j - x^i}{\mu} = \frac{1}{|\mathcal{E}|} \sum_{(i,j) \in \mathcal{E}} (\xi_j - \xi_i). \tag{3}$$

Then the point can be updated with $x_{\text{new}} = x - \eta \tilde{g}(x)$, where $\eta$ is the learning rate and $\tilde{g}(x)$ is the estimated gradient. With the help of preference ranking oracle and ZO-RankSGD algorithm, we are able to optimize the prompt guiding the agent towards human preferences in the target environment.

## 4 PROMPT-TUNING DECISION TRANSFORMER

This section introduces prompt-tuning as a memory-efficient alternative to full-model fine-tuning for the pre-trained agents in the context of few-shot policy generalization tasks. We begin by presenting the problem formulation in Section 4.1 and subsequently provide a formal definition of our method in Section 4.2. The overall procedure of our proposed Prompt-Tuning DT is illustrated in Figure 1.

### 4.1 PROBLEM FORMULATION

In our few-shot evaluation experiments, our objective is to align the output of the PLM with human preferences using a restricted number of offline trajectories and limited oracle calls, all accomplished

in a parameter-efficient manner. To better quantitatively evaluate our method, we adopt *high cumulative reward*, a widely-used indicator in the field of RL, as a representation of human preference and conduct experiments on few-shot generalization tasks, which involve training the agent on a set of tasks using offline data and evaluating its ability to generalize to new tasks.

There are two distinct sets of tasks, denoted as $\mathcal{T}^{train}$ and $\mathcal{T}^{test}$, ensuring that there is no overlap between them ($\mathcal{T}^{train} \cap \mathcal{T}^{test} = \emptyset$). This arrangement requires the model to perform well on tasks with goals that lie outside the training range, thereby necessitating the ability to generalize to out-of-distribution tasks. Each task $\mathcal{T}_i$ in the training set $\mathcal{T}^{train}$ is associated with a corresponding dataset $\mathcal{D}_i$, which consists of pre-collected trajectories obtained using an unknown behavior policy $\pi_i$. For a test task $\mathcal{T}_i \in \mathcal{T}^{test}$, there are two possible approaches to adapt to the new domain. One approach involves updating the model parameters using task-specific offline data $\mathcal{P}_i$, which is usually much less than the training dataset $|\mathcal{P}_i| << |\mathcal{D}_i|$. Alternatively, one can incorporate task-specific prompts derived from $\mathcal{P}_i$ to mitigate the issue of distribution shift, although such approaches are generally considered inferior to fine-tuning methods (Brown et al., 2020). Our method combines the advantages of both approaches to fine-tune the prompts.

---

**Algorithm 1** Prompt-Tuning DT

---

**Require:** Initial prompt $\tau_0^*$, stepsize $\eta$, iterations $T$, smoothing parameter $\mu$.
1: Construct the initial point from prompt: $x_0 = \hat{r}_1^* \,||\, s_1^* \,||\, a_1^* \,||\, \ldots \,||\, \hat{r}_{K^*}^* \,||\, s_{K^*}^* \,||\, a_{K^*}^*$.
2: **for** $t = 1$ to $T$ **do**
3:     Sample $m$ i.i.d. random vectors $\{\xi_{(t,1)}, \cdots, \xi_{(t,m)}\}$ from $N(0, I_{d_x})$.
4:     Query the ranking function to obtain the exact value with offline loss function 6 or online reward function 7 with input $\{x_{t-1} + \mu\xi_{(t,1)}, \cdots, x_{t-1} + \mu\xi_{(t,m)}\}$.
5:     Construct the corresponding DAG $\mathcal{G} = (\mathcal{N}, \mathcal{E})$ as described in Section 3.2.
6:     Compute the gradient estimator using: $g_t = \frac{1}{|\mathcal{E}|} \sum_{(i,j) \in \mathcal{E}} (\xi_{(t,j)} - \xi_{(t,i)})$.
7:     $x_t = x_{t-1} - \eta g_t$.
8: **end for**

---

## 4.2 DEEP BLACK-BOX TUNING

Trajectory prompts contain only the necessary information to aid in task identification while being insufficient for the agent to imitate, thus the length of the prompt $K^*$ is usually less than 10. Each trajectory prompt contains multiple tuples of state $s^*$, action $a^*$ and reward-to-go $\hat{r}^*$, denoted as $(s^*, a^*, \hat{r}^*)$, following the representation in (Chen et al., 2021; Xu et al., 2022). Each element with superscript $\cdot^*$ is associated with the trajectory prompt, which can be formulated as:

$$\tau^* = (\hat{r}_1^*, s_1^*, a_1^*, \ldots, \hat{r}_{K^*}^*, s_{K^*}^*, a_{K^*}^*) \tag{4}$$

In contrast to the prompt learning approach typically employed in NLP, where a cloze test-style textual prompt is presented to the model for filling in the corresponding answer, the trajectory prompt utilized in the Decision Transformer consists of tokens that have unique representations and physical interpretations. These tokens are carefully crafted to represent essential components of RL tasks, including the state, action, and return-to-go. The state token encapsulates the environmental information of the agent at a given position and is usually represented by a high-dimensional vector. On the other hand, the action token exhibits significant variations across dimensions, with specific values corresponding to distinct actions. Moreover, the return-to-go token serves to denote the expected reward that we aim for the agent to attain. Given these distinct characteristics of RL prompts, directly applying prompt-tuning approaches from NLP becomes challenging: RL prompts are specifically tailored to guide agent behavior by leveraging environmental modeling and analysis, rather than primarily focusing on adjusting the prompt format as in NLP prompt learning.

We utilize the ZO-RankSGD optimization approach to update the trajectory prompt. This method avoids explicit gradient computation and eliminates the necessity for intricate understanding of the particular structure of the PLM. Given the initial trajectory prompt $\tau_0^*$, we concatenate one trajectory segment as a unit and add a standard Gaussian distribution to it to avoid catastrophic deviations:

$$\begin{aligned} x_0 &= \hat{r}_1^* \,||\, s_1^* \,||\, a_1^* \,||\, \ldots \,||\, \hat{r}_{K^*}^* \,||\, s_{K^*}^* \,||\, a_{K^*}^*, \\ x_0^i &= x_0 + \mu\xi_i, \ \ \xi_i \sim \mathcal{N}(0, I_{d_x}), \end{aligned} \tag{5}$$

where $||$ means concatenation, $\hat{r}_i^* \in \mathbb{R}^{d_r}, s_i^* \in \mathbb{R}^{d_s}, a_i^* \in \mathbb{R}^{d_a}$, and $d_x = (d_r + d_s + d_a) \times K^*$.

For the ranking function $f$, we propose two preference ranking functions tailored to different RL environments (offline and online): offline loss function and online reward function. For the offline setting, where we have access to a set of trajectories $\mathcal{P}$ collected in advance, we utilize the MSE loss between the true action and predicted action as the preference ranking function:

$$f(x_0^i) = \mathbb{E}_{\tau^{\text{offline}} \sim \mathcal{P}} \left[ \frac{1}{T} \sum_{t=1}^{T} (a_t - \pi(x_0^i, \tau_t^{\text{offline}}))^2 \right]. \tag{6}$$

While for the online setting, where we can interact with the simulator, we consider the episode accumulated reward obtained by the model during online interactions as the preference ranking function, which is represented as follows:

$$f(x_0^i) = -\mathbb{E}_{\tau^{\text{online}}} \left[ \sum_{t=1}^{\infty} \mathcal{R}(s_t, \pi(x_0^i, \tau_t^{\text{online}})) \right]. \tag{7}$$

Note that the function is optimized to the minimal, we need to add a minus sign in front of the equation 7. In both cases, the selection of the preference ranking function aims to adapt to the human preference for high cumulative reward, which also serves as a widely used metric for evaluating a pre-trained model's performance. Then the ranking oracle $O_f^{(m,k)}$ simply returns the order of these values which is subsequently utilized for computing the gradient estimator.

Human judgment can also be employed as an oracle to rank these trajectories based on individual preferences. However, this study does not delve into comprehensive experiments within human-in-the-loop settings, leaving this aspect for future investigations. In this context, we primarily showcase the algorithm's potential in a human-in-the-loop framework from a design perspective: (1) Ranking information possesses a unique appeal to humans as it offers a straightforward and intuitive means to express personal preferences without the need for exact scores or ratings, making our approach user-friendly. (2) The forward-forward fine-tuning strategy proves advantageous in terms of conserving GPU memory, which bears significance for deployment on devices with limited resources. (3) Ranking-based approaches avoid intricate understanding of the PLM's structure, where leveraging the PLM's API enables effective prompt fine-tuning in alignment with human preferences. Collectively, these attributes render our method well-suited for human-in-the-loop environments. The primary objective of this article is to establish the method's feasibility and efficiency.

We summarize the entire procedure of prompt-tuning in Algorithm 1. Prompt-Tuning DT employs an approximate gradient calculation to adapt the pre-trained agent to specific preferences. Gaussian noise is introduced to the initial prompt, driving Prompt-Tuning DT to discover a more expressive prompt tailored to the target tasks. There are two options available for the ranking function. The offline loss function requires pre-collected datasets from the target tasks in $\mathcal{T}^{test}$, while the online reward function assumes interaction with a simulator for the target tasks in $\mathcal{T}^{test}$. After $T$ iterations of the fine-tuning process, we utilize the optimized result to initialize the prompt $\tau^*$ at the onset of the evaluation stage and update the recent history $\tau$ with streamingly collected data.

## 5 EXPERIMENT

We perform experiments to assess the aligning human preference of Prompt-Tuning DT by using the episode accumulated reward as the evaluation metric. Our experimental evaluation seeks to answer the following research questions: (1) Can the prompt-tuning approach achieve comparable performance to full-model fine-tuning with limited ranking oracle calls? (2) What is the impact of the fine-tuning dataset size on the effectiveness of the prompt-tuning approach? (3) How does the quality and quantity of the prompt influence the effectiveness of the prompt-tuning approach? (4) How does the hyper-parameter influence the effectiveness of the prompt-tuning approach?

### 5.1 DATASETS AND TASKS

In this study, we assess the performance of our proposed approach on several datasets that are used in meta-RL (Finn et al., 2017; Rothfuss et al., 2018; Mitchell et al., 2021; Yu et al., 2020a), namely Cheetah-dir, Cheetah-vel, Ant-dir, and Meta-World reach-v2. The objectives of these tasks are

Table 1: Results for meta-RL control tasks. The best mean scores are highlighted in bold. Each environment has prompts of length $K^* = 5$ and limited fine-tuned datasets. Scores are normalized so that 100 represents an expert policy. Notably, our methods outperform other parameter-efficient methods on almost all tasks and even achieve comparable performance with Full-Model-FT method.

| Environment | PLM | PDT | Soft-Prompt | Adaptor | PTDT-offline | PTDT-online | Full-Model-FT |
|---|---|---|---|---|---|---|---|
| **Random Prompt Initialization** | | | | | | | |
| Cheetah-dir | -3.8 ± 0.3 | 94.7 ± 0.0 | **95.6 ± 0.3** | 74.5 ± 2.0 | 95.5 ± 0.0 | 95.1 ± 0.6 | 93.3 ± 1.0 |
| Cheetah-vel | 7.1 ± 0.9 | 44.2 ± 0.1 | 44.6 ± 0.1 | 19.7 ± 4.6 | **61.2 ± 3.2** | 60.5 ± 7.9 | 44.0 ± 9.8 |
| Ant-dir | 24.7 ± 1.4 | 61.7 ± 0.2 | 66.5 ± 0.4 | 75.3 ± 6.6 | 75.3 ± 5.9 | **78.7 ± 0.2** | 77.5 ± 1.7 |
| MW reach-v2 | 45.5 ± 1.0 | 42.1 ± 5.8 | 41.8 ± 5.8 | 0.3 ± 0.1 | **54.0 ± 2.2** | 49.9 ± 4.4 | 43.9 ± 13.0 |
| **Average** | 18.4 | 60.7 | 62.1 | 41.2 | **71.5** | 71.0 | 64.7 |
| **Expert Prompt Initialization** | | | | | | | |
| Cheetah-dir | -3.8 ± 0.3 | 94.6 ± 0.5 | **95.6 ± 0.1** | 75.5 ± 0.9 | 95.5 ± 0.1 | 95.4 ± 0.1 | 93.6 ± 0.7 |
| Cheetah-vel | 7.1 ± 0.9 | 86.0 ± 1.4 | 86.4 ± 1.2 | 27.1 ± 4.5 | **87.8 ± 0.4** | 87.5 ± 0.1 | 81.9 ± 0.5 |
| Ant-dir | 24.7 ± 1.4 | 71.3 ± 0.3 | 72.7 ± 0.3 | 76.8 ± 6.6 | 75.5 ± 0.4 | 74.5 ± 0.3 | **84.2 ± 1.6** |
| MW reach-v2 | 45.5 ± 1.0 | 50.9 ± 6.6 | 50.9 ± 6.6 | 0.3 ± 0.1 | **56.5 ± 1.0** | 53.1 ± 3.0 | 54.2 ± 8.6 |
| **Average** | 18.4 | 75.7 | 76.4 | 43.1 | **78.8** | 77.6 | 78.5 |

distinct, with Cheetah-dir and Ant-dir incentivizing high velocity in the goal direction, Cheetah-vel penalizing deviations from the target velocity using $l_2$ errors, and Meta-World reach-v2 task requiring the robot's end-effector to reach a designated position in 3D space. These tasks require the agent to move forward as far and quickly as possible. Detailed environment and hyper-parameter settings refer to Appendix A and B.

We adopt the dataset construction and settings from (Mitchell et al., 2021) for the meta-RL control tasks considered in this study. Specifically, the datasets comprise the full replay buffer of Soft Actor-Critic (Haarnoja et al., 2018) for Cheetah-dir and Ant-dir, and TD3 (Fujimoto et al., 2018) for Cheetah-vel. Expert trajectories are collected for Meta-World reach-v2 (Yu et al., 2020a) using script expert policies provided in the environment.

## 5.2 BASELINES

We assess the few-shot generalization capabilities of Prompt-Tuning DT by comparing it against five baseline methods across meta-RL control tasks. Our approach begins with pre-training a PLM on the training tasks $\mathcal{T}^{train}$ using the DT architecture. Subsequently, this PLM is directly applied for inference on the test tasks $\mathcal{T}^{test}$ without any additional operations, denoted as "PLM" in Table 1. We then explore four additional few-shot learning methods: (1) PDT (Xu et al., 2022) involves the collection of prompts from the target tasks $\mathcal{T}^{test}$ and incorporates these prompts into the input to aid the PLM in better adapting to the target tasks, which is known as straightforward prompt-based adaptation. (2) Soft-Prompt treats these collected prompts as soft embeddings and employs a separate optimizer to fine-tune these embeddings, akin to prevalent practices in the NLP domain (Lester et al., 2021). (3) Adaptor refers to a widely-used parameter-efficient technique previously explored in HDT (Xu et al., 2023). We integrate adaptors, following HDT's methodology, into each decoder module of the PLM. During inference, only the adaptors are fine-tuned to enhance the PLM's adaptation to the target tasks. (4) We also consider the full-model fine-tuning method, which serves as an upper performance bound for fine-tuning techniques in scenarios with complete data (Li & Liang, 2021). This breadth allows for a thorough evaluation of the efficacy of our methods.

Our approach encompasses two distinct variations: Prompt-Tuning DT with an offline loss function (PTDT-offline) and Prompt-Tuning DT with an online reward function (PTDT-online). To maintain fairness in the comparison, all offline methodologies are confined to utilizing an equivalent quantity of samples $\mathcal{P}_i$ sourced from the target task $\mathcal{T}^{test}$. While PTDT-online involves interaction with a simulator, we meticulously regulate the number of interactions to guarantee access to new trajectories of comparable magnitudes. Note that all methods utilize the same PLM to ensure an equitable comparison. By aligning these experimental setups, we establish a robust foundation for an unbiased assessment of the method's performance, thereby enhancing the validity of our findings.

## 5.3 MAIN RESULTS

We perform a comparative analysis between Prompt-Tuning DT and the parameter-efficient baseline methods to assess their few-shot generalization abilities and evaluate the tuning efficiency of

Prompt-Tuning DT in relation to the full-model fine-tuning approach. We use the episode accumulated reward as the evaluation metric in the test task set $\mathcal{T}^{test}$. Note that we present two prompt initialization settings: the random prompt, gathered from a random policy, and the expert prompt, acquired from the expert policy. The main results are normalized and presented in Table 1, which showcases the few-shot performance of various algorithms (more details are presented in Appendix).

The outcomes from the PLM underscore the significance of prompts, as PLM struggles to adapt to target tasks in zero-shot scenarios. During the random prompt initialization setting, PDT effectively leverages pre-collected prompts by incorporating them into PLM input, resulting in substantial improvements. Adaptor also exhibits enhanced performance over PLM by introducing supplementary fine-tuning adaptors within decoder modules. However, its efficacy is hampered, particularly in the MW reach-v2 and cheetah-vel environments, likely due to limited target datasets $\mathcal{P}_i$. Both Soft-Prompt and our proposed approach undertake further fine-tuning of the prompt itself. While Soft-Prompt treats the prompt as an embedding and optimizes it using the AdamW optimizer, it achieves better results than PDT but lags behind our approach. This discrepancy can be attributed to the intricate and environment-specific nature of RL prompts, which necessitate meticulous updates (visualized in Appendix C). Our approach demonstrates effectiveness in both offline and online settings, yielding notable performance improvements that surpass the benchmark of Full-Model-FT, which serves as a primary reference for evaluating the efficacy of our approach.

Under the expert prompt initialization setting, characterized by strong prior knowledge, all baseline approaches exhibit substantial enhancements in comparison to their counterparts in the random initialization setting. Moreover, the relative improvement of our method, compared to other approaches, diminishes under the expert prompt regime. This reduction can be attributed to the strong prior condition introduced by expert trajectories, which constrains the extent of improvement across methods. Nevertheless, despite this limitation, our approach consistently outperforms all other parameter-efficient methods. Furthermore, our method achieves outcomes on par with Full-Model-FT. Collectively, these outcomes accentuate the effectiveness of our proposed prompt-tuning techniques across both random and expert prompt initialization scenarios.

## 5.4 ABLATION

**Random Search.** Considering the random search could lead to a lot of variability in the performance of the algorithm, we further investigate the impact of the number $m$ and variance $\mu$ of Gaussian random variables. The results are shown in Figure 2(a). When we increase the number of samples $m$ during each update, the algorithm explores a larger set of possible directions to evaluate the performance, leading to a more accurate gradient estimation (variance decrease). However, as $m$ increases, the burden on the oracle, which needs to provide ranking information for the $m$ samples, also grows. On the other hand, increasing the variance of the Gaussian distribution $\mu$ allows the algorithm to explore a broader range of potential directions for performance evaluation. However, a higher variance of the Gaussian $\mu$ also introduces larger variability in the gradient estimation, which may not consistently guarantee performance improvement.

**Prompt Length.** We investigate the impact of prompt length on the prompt-tuning methods, considering its influence on both the number of tuning parameters in the approach and the speed of inference. It is crucial to strike a balance between the richness of information provided by the prompt and the effectiveness of the prompt-tuning process. The results are shown in Figure 2(b). Our primary focus is on investigating Soft-Prompt and PTDT-offline. We employ PDT as the baseline, which does not involve additional fine-tuning processes. The augmentation of the prompt does not uniformly lead to enhanced generalization performance for both PDT and Soft-Prompt. This trend underscores the efficacy of our approach in adeptly refining prompts, even when confronted with a greater number of tuning parameters.

**Sample Efficiency.** We explore the impact of progressively increasing the number of fine-tuning samples on the performance of fine-tuning approaches, aiming to understand the prompt-tuning methods' dependence on the quantity of fine-tuning samples. Figure 3 illustrates the performance trends of these methods on the Cheetah-dir, Cheetah-vel, Ant-dir, and MetaWorld-reach-v2 environments. Prompt-tuning methods (PTDT, Soft-Prompt) exhibit consistent performance across varying sample sizes, whereas model-tuning methods (Adaptor, Full-Model-FT) exhibit incremental improvements as the number of samples increases.

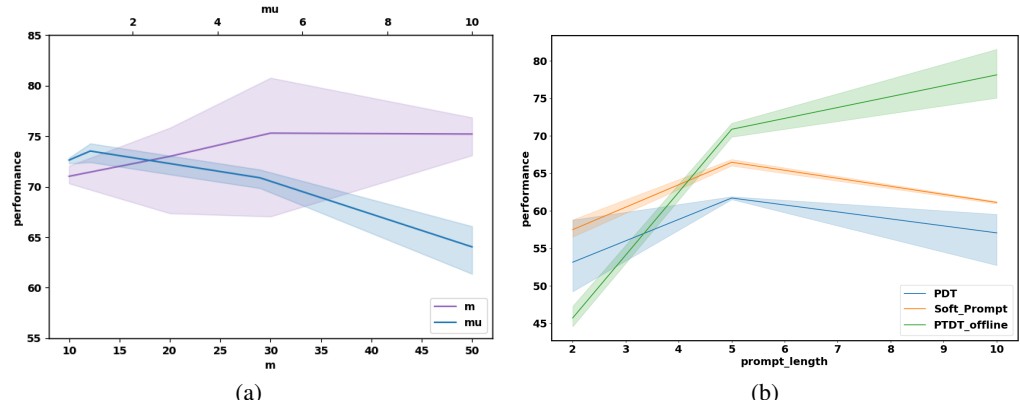

Figure 2: (a) Ablation analysis concerning the influence of parameters $m$ and $\mu$ for our PTDT-offline approach. (b) Ablation investigation into the effect of prompt length for the prompt-tuning methodologies. The experiments are conducted within the Ant-dir environment, and the reported results are averaged across three independent seeds, with normalization applied.

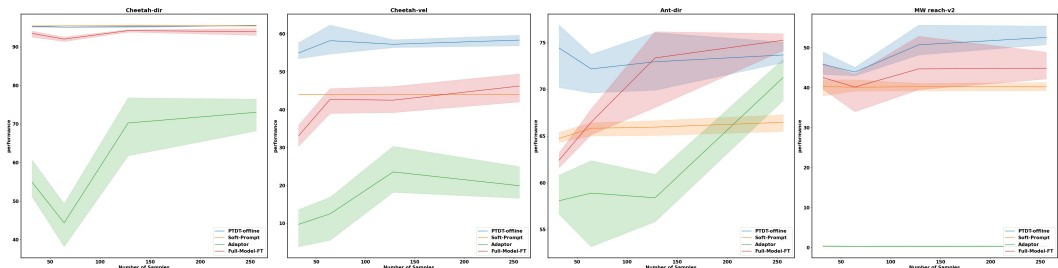

Figure 3: Comparison between different fine-tuning approaches when different numbers of training samples are available. The figures from left to right correspond to Cheetah-dir, Cheetah-vel, Ant-dir, and MetaWorld-reach-v2 environments respectively. Each plot is run with 3 seeds. The x-axis is the training size and the y-axis is the evaluation metric (higher is better).

Furthermore, it is worth noting that unlike the observed phenomenon in NLP (Li & Liang, 2021; Gu et al., 2021), the performance of the these fine-tuning approaches does not exhibit a monotonically increasing trend as the amount of fine-tuning data increases. In all environments, a downward inflection point is observed in the performance curve as the number of samples increases. This can be attributed to the presence of "bad samples" in the training dataset, which adversely impact the fine-tuning process and potentially result in catastrophic deviations. By utilizing a larger dataset, the proportion of "bad samples" decreases, enabling the fine-tuning to converge to more optimal policies and improve overall performance.

## 6 CONCLUSION

In this paper, we introduce Prompt-Tuning Decision Transformer (Prompt-Tuning DT), a novel algorithm that aligns with human preferences in the target environment. By optimizing prompts without back-propagation, Prompt-Tuning DT offers a memory-efficient alternative to fine-tuning PLMs. Furthermore, our prompt-tuning offline RL framework using trajectory prompts allows for effective adaptation to new tasks with minimal parameter optimization and a small number of trajectories. Through extensive experiments, our approach achieves performance on par with full-model fine-tuning and surpasses alternative parameter-efficient methods.

Our work contributes to the advancement of prompt-tuning approaches in RL, providing a promising direction for optimizing pre-trained large-scale RL agents for specific preferences and downstream tasks. Our approach demonstrates the potential of prompt-tuning methods in RL settings and opens up avenues for future research in developing tailored prompt-tuning techniques for RL agents. We envision that prompt-tuning approaches will continue to play a crucial role in enhancing the generalization and adaptability of RL agents in real-world scenarios.

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

# Appendices

## A  DETAILED ENVIRONMENT

We evaluate our approach on a variety of tasks, including meta-RL control tasks. These tasks can be described as follows:

- Cheetah-dir: The task comprises two directions: forward and backward, in which the cheetah agent is incentivized to attain high velocity along the designated direction. Both the training and testing sets encompass these two tasks, providing comprehensive coverage of the agent's performance.

- Cheetah-vel: In this task, a total of 40 distinct tasks are defined, each characterized by a different goal velocity. The target velocities are uniformly sampled from the range of 0 to 3. The agent is subjected to a penalty based on the $l_2$ error between its achieved velocity and the target velocity. We reserve 5 tasks for testing purposes and allocate the remaining 35 tasks for training.

- Ant-dir: There are 50 tasks in Ant-dir, where the goal directions are uniformly sampled in a 2D space. The 8-joint ant agent is rewarded for achieving high velocity along the specified goal direction. We select 5 tasks for testing and use the remaining tasks for training.

- Meta-World reach-v2: This task involves controlling a Sawyer robot's end-effector to reach a target position in 3D space. The agent directly controls the XYZ location of the end-effector, and each task has a different goal position. We train on 15 tasks and test on 5 tasks.

By evaluating our approach on these diverse tasks, we can assess its performance and generalization capabilities across different control scenarios.

The generalization capability of our approach is evaluated by examining the task index of the training and testing sets, as shown in Table 2. The experimental setup in Section 5 adheres to the training and testing division specified in Table 2. This ensures consistency and allows for a comprehensive assessment of the approach's performance across different tasks.

Table 2: Training and testing task indexes when testing the generalization ability in meta-RL tasks

| | |
|---|---|
| **Cheetah-dir** | |
| Training set of size 2 | [0,1] |
| Testing set of size 2 | [0.1] |
| **Cheetah-vel** | |
| Training set of size 35 | [0-1,3-6,8-14,16-22,24-25,27-39] |
| Testing set of size 5 | [2,7,15,23,26] |
| **ant-dir** | |
| Training set of size 45 | [0-5,7-16,18-22,24-29,31-40,42-49] |
| Testing set of size 5 | [6,17,23,30,41] |
| **Meta-World reach-v2** | |
| Training set of size 15 | [1-5,7,8,10-14,17-19] |
| Testing set of size 5 | [6,9,15,16,20] |

## B  HYPERPARAMETERS

We show the hyperparameters of Prompt-Tuning DT in Table 3.

Table 3: Common Hyperparameters of Prompt-Tuning DT.

| Hyperparameters | Value |
|---|---|
| $K$ (length of context $\tau$) | 20 |
| training batch size for each task | 32 |
| training number of steps per iteration | 10 |
| training max iterations | 5000 |
| fine-tuning batch size for each task | 32 |
| fine-tuning number of steps per iteration (offline) | 50 |
| fine-tuning number of steps per iteration (online) | 10 |
| fine-tuning max iterations | 20 |
| number of evaluation episodes for each task | 50 |
| learning rate | [1e-3, 1e-6, 1e-8] |
| learning rate decay weight | 1e-4 |
| ranking algorithm m | [10, 20, 30, 50] |
| ranking algorithm $\mu$ | [0.5, 1.0, 5.0, 10.0] |
| Return-to-go conditioning | 1500 Cheetah-dir |
| | 0 Cheetah-vel |
| | 500 Ant-dir |
| | 650 Meta-World reach-v2 |
| Pompt length $K^*$ | 5 |
| number of layers | 3 |
| number of attention heads | 1 |
| embedding dimension | 128 |
| activation | ReLU |

## C  ABLATION STUDY

In this section, we provide additional visual supplements to enhance the intuitiveness of our experimental results in the ablation study. These supplementary visuals aim to provide a clearer representation of our findings and further support our conclusions.

We generate heat maps to compare the original prompt (samples 0-4) with the prompt after fine-tuning (samples 5-9) in Figure 4. The heat maps illustrate significant differences in certain dimensions resulting from prompt-tuning. Specifically, when examining the State Vector, we observe a consistent trend of low values in dimensions such as the 7th and 15th. This pattern is also observed in the Action Vector and Return-to-go Vector. While Figure 4 itself may not provide directly interpretable content, it offers insights into the dimensions that play a critical role in determining the final performance.

Our primary goal is to ensure that the prompts retain their physical meaning while effectively converging to the optimal trajectory prompt, thereby benefiting downstream tasks. Recognizing that presenting valid information in high-dimensional space can be challenging, we leverage the t-SNE method to reduce the dimensionality of the data and visualize the final results in a two-dimensional graph. This approach offers a more intuitive representation of the prompt-tuning process and allows for a better interpretation of the learned prompt in a reduced space. We primarily present three categories of prompts: the original prompt, along with two prompt-tuning methods, namely Soft Prompt and our proposed approach PTDT.

As illustrated in Figure 5, the prompts obtained through our approach maintain their structural characteristics, resembling the original prompts. On the other hand, when fine-tuning prompts using soft prompts, the resulting prompts lose their physical meaning and appear as mere embeddings that may only partially contribute to the final performance. This phenomenon is clearly discernible in the visualization, which exhibits a considerable deviation from the original prompt.

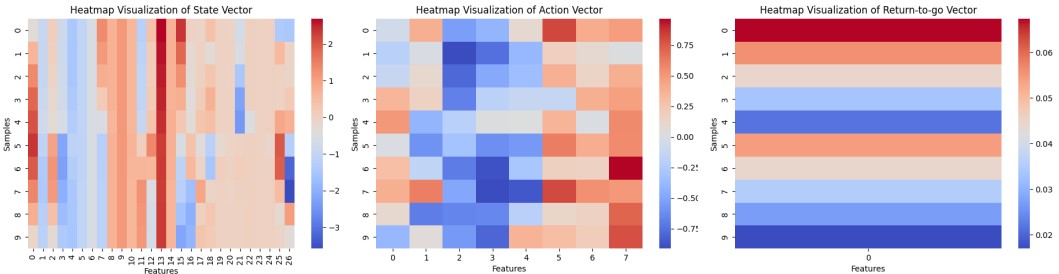

Figure 4: Ablation on the visualization of prompts. For the visualization, we generate State Vectors, Action Vectors, and Return-to-go Vectors from 10 different samples. Samples 0-4 represent the initial prompts, while samples 5-9 represent the prompts after fine-tuning. The results provide valuable insights into the dimensions that significantly influence the final performance.

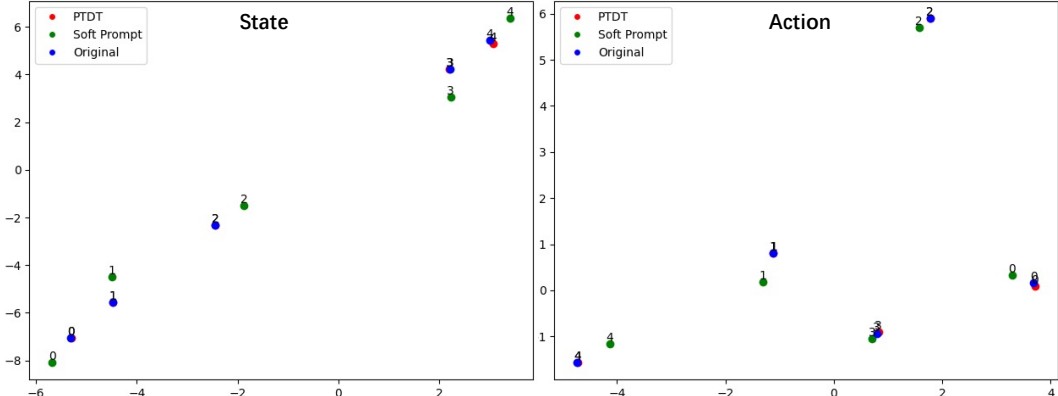

Figure 5: Ablation on the visualization of prompts. For the visualization, we use the t-SNE method to reduce the dimensionality of the data and visualize the final results in a two-dimensional graph.

## D    NORMALIZATION

Table 5 presents the normalized scores used for normalization. The values are selected from the collected datasets, with "Random" corresponding to the minimal value and "Gamer" corresponding to the maximal value.

Table 4: Baseline scores used for normalization.

| Game | Random | Gamer |
|------|--------|-------|
| Cheetah-dir | $-10.1$ | 994.73 |
| Cheetah-vel | $-200.018$ | $-18.864$ |
| Ant-dir | $-31.046$ | 521.31 |
| Meta-World Reach-v2 | 0 | 555.01 |

## E    MORE ANALYSIS ABOUT RESULTS

Due to space constraints, we have included a portion of the results analysis in the appendix, specifically for Table 1.

When contrasting random and expert initialization conditions, it is crucial to highlight that the ultimate performance of our PTDT-offline method, when initialized randomly, has not only matched but in certain instances, exceeded the outcomes attained through expert initialization (in comparison

with PDT-expert). This phenomenon can be attributed to the high efficiency of our method, which is capable of rapidly steering randomly initialized prompts into the proximity of expert-level prompts.

Furthermore, in the context of the PTDT-online method, there is an intriguing observation: the experimental outcomes in the Ant-dir environment, when initialized randomly, surpass those achieved through expert initialization. This counterintuitive finding warrants attention and further analysis. We attribute this phenomenon to the presence of local optima when initialized with expert prompts. Specifically, when the prompt is randomly initialized, our algorithm correctly estimates the convergence direction, allowing for more effective convergence toward a global optimum at a larger learning rate. Conversely, when the prompt is initialized with expert guidance, it is already situated within a local optima region. Therefore, if learning continues with a larger learning rate, its performance gradually deteriorates, moving away from the local convergence region. Conversely, if learning continues with a smaller learning rate, it remains constrained within the local convergence domain. Hence, it is possible that the performance of PTDT-online under random initialization conditions might surpass its performance under expert initialization conditions.

However, this pattern is not observed for PTDT-offline. This discrepancy can be attributed to the fact that PTDT-online can directly interact with the simulator, and the cumulative rewards it acquires provide unbiased estimates of the current policy, facilitating the accurate estimation of the global convergence direction. In contrast, PTDT-offline is constrained by the limitations of the offline dataset, making it unable to achieve unbiased estimation of the global convergence direction.

## F  PARAMETER SIZE

We also present the number of parameters that each method requires for updates during the fine-tuning phase. Notably, our prompt-tuning method demonstrates the capability to achieve superior performance while necessitating updates to only a very small fraction of parameters.

Table 5: The number of parameters required for each method and corresponding results.

| Algorithms | PLM | PDT | Soft-Prompt | Adaptor | PTDT-offline | PTDT-online | Full-Model-FT |
|---|---|---|---|---|---|---|---|
| Parameter Size | 0 | 0 | 0.24K | 13.49K | 0.24K | 0.24K | 0.87M |
| Random Setting | 18.4 | 60.7 | 62.1 | 41.2 | 71.5 | 71.0 | 64.7 |
| Expert Setting | 18.4 | 75.7 | 76.4 | 43.1 | 78.8 | 77.6 | 78.5 |

