# OpenReview forum: "Prompt-Tuning Decision Transformer with Preference Ranking"
_ICLR.cc/2024/Conference — ICLR 2024 Conference Withdrawn Submission_

### Official Review · Reviewer_2gif · 2023-10-25

**Soundness:** 2 fair
**Presentation:** 2 fair
**Contribution:** 2 fair
**Rating:** 5
**Confidence:** 4

**Summary:**

The paper introduces a novel black-box tuning approach, focusing on optimizing prompts for the Decision Transformer (DT) within the context of offline meta-reinforcement learning. Drawing inspiration from prompt learning methodologies prevalent in Natural Language Processing (NLP), the strategy involves fine-tuning the prompt segment specifically. This fine-tuning utilizes task-specific offline data, typically scarce compared to conventional training datasets, aiming to achieve specific goals: minimizing behavior cloning loss in offline settings or maximizing returns in online contexts. Even when the tuning process is confined to the minimal parameters essential for prompt adjustment, the proposed method demonstrates performance on par with comprehensive model fine-tuning in scenarios with limited data availability (few-shot settings).

**Strengths:**

**Strength 1: Plausible Application of Prompt-Tuning in RL**

The introduction of prompt-tuning to the realm of Reinforcement Learning (RL) and specifically its integration with the Decision Transformer stands out as a formidable strength of this work. The use of a black-box tuning methodology, coupled with a ranking optimization framework for prompt tuning, is particularly noteworthy. This approach avoids the necessity for a comprehensive, end-to-end gradient update of the entire model, presenting a novel, convincing, and computationally efficient strategy. Also the empirical result with random prompt initialization suffices to prove the efficacy of the proposed method.

**Strength 2: Comprehensive Ablation Studies**

The paper presents thorough and insightful ablation studies, offering a detailed analysis that contributes significantly to understanding the proposed method's effectiveness. A standout aspect is the content presented in Figure 5 of the appendix, which emphasizes the critical role and impact of the prompt-tuning process.

**Weaknesses:**

**Weakness 1: Need for Greater Clarity in Problem Formulation**

This work delves into a crucial application within the domain of Reinforcement Learning (RL), yet there is room for enhancing the clarity of the problem formulation, specifically concerning the concepts of 'meta' and 'offline' in the context of offline meta-RL.

- **Meta Aspect**: The manuscript appears to presuppose the availability of an appropriate, task-specific prompt at the outset of each task. However, the essence of meta-RL traditionally entails a mechanism to infer the dynamics of the current task [1]. This distinction is crucial, as the described setup leans more toward a 'multi-task setup,' assuming a known index for every task (May be the random prompt initialization does this work?). While prior works such as MACAW  [2] and Prompt-DT [3] also frame their contributions in the context of 'meta-learning,' there seems to be a slight deviation from standard expectations. There's a risk of misinterpretation, potentially leading readers to assume that the prompt evolves independently from a task-independent prompt, given an unidentified task index.

- **Offline Aspect**: The scope of the research extends beyond purely offline methodologies, especially considering the inclusion of reward-based tuning as seen in Eq. 7. Consequently, it would be appropriate to broaden the scope of the related work section to encompass studies that bridge both online and offline setups, including references such as [4,5]. While the current method maintains considerable benefits due to its limited use of steps and focus on prompt tuning, acknowledging and discussing related works would help situate this study within the broader scholarly discourse. It is essential for readers to understand the specific advantages and novel positioning of this work compared to these related methodologies.


**Weakness 2: Enhancements Needed in Empirical Validation**

While this research primarily adopts an empirical approach, there are areas where the robustness of empirical validation could be fortified. The absence of source code is a noticeable limitation that adds a layer of complexity in grasping the finer nuances of the implementation, particularly the operational specifics of the prompt training and initialization. The details in the appendix could benefit from further expansion to alleviate this concern.

Moreover, in the realm of meta-reinforcement learning, the consistency and reliability of results are paramount. The current practice of utilizing three random seeds for experiments, although commonly accepted, may not suffice to capture the full variability and confirm the stability of the proposed method. It is advisable for the authors to consider expanding the number of random seeds to reinforce the statistical significance of the results, enhancing confidence in the findings' replicability and robustness.

Also there are some questions regarding the experiments (see the Questions below)

(Minor Points)

In Equation 1, maybe $a_t$ should be removed.

For Figure 3, increasing the font size would greatly enhance readability and comprehension.

Certain notational choices in the paper could benefit from reconsideration to enhance clarity. For instance, using "\mu" to signify the standard deviation may lead to confusion, as this symbol often denotes the mean or a behavior policy in statistical contexts. Additionally, the use of "||" for concatenation in Equation 5 may be misleading, given its common interpretation as a norm notation in mathematical literature.

[1] Beck et al., A Survey of Meta-Reinforcement Learning, 2023.

[2] Mitchell et al., Offline Meta-Reinforcement Learning with Advantage Weighting, 2021.

[3] Xu et al., Prompting Decision Transformer for Few-Shot Policy Generalization, 2022.

[4] Lee et al., Offline-to-Online Reinforcement Learning via Balanced Replay and Pessimistic Q-Ensemble, 2021.

[5] Zheng et al., Online Decision Transformer, 2022.

**Questions:**

Question 1:

The observation that PTDT-online does not surpass its offline counterpart, despite utilizing additional online samples, raises questions regarding the underlying factors. Could the discrepancy in performance be attributed to the distinct objectives outlined in equations 6 and 7? If this is the case, the rationale behind proposing PTDT-online warrants further clarification, as one might expect the incorporation of online data to enhance, rather than diminish, the model's efficacy.

Question 2:

Table 1 presents an intriguing scenario where PTDT-offline outperforms PDT with an expert prompt in nearly all environments, with the exception of cheetah-vel. This outcome prompts several questions. Is it possible that PTDT, even when initiated with a random prompt, evolves to generate a prompt on par with or superior to that constructed by an expert? Alternatively, could this suggest that the supposed advantage of an expert prompt does not universally translate across all environments? Understanding the dynamics that led to these results could provide valuable insights into the adaptability and utility of prompts in varying contexts.

---

### Official Review · Reviewer_Yeg6 · 2023-10-30

**Soundness:** 2 fair
**Presentation:** 3 good
**Contribution:** 2 fair
**Rating:** 5
**Confidence:** 4

**Summary:**

This paper presents Prompt-Tuning Decision Transformer (Prompt-DT), which extends transformer models to offline Reinforcement Learning tasks. This method utilizes trajectory prompts that allow few-shot generalization to unseen tasks. This design allows the agent to adapt quickly to new environments or situations using minimal examples. This work also introduces a ranking optimization technique that doesn't require explicit gradient information but uses rank-based data to optimize the model. This method is combined with a rank-based gradient estimator. The goal of this technique is to guide agents towards human preferences in their target environments. This is especially important in contexts where human preferences are critical, like healthcare or finance.

**Strengths:**

1. This paper proposes a black-box optimized prompt-tuning decision transformer method, which can quickly generalize to unseen tasks as shown in the evaluation results.

2. This paper utilizes rank-based optimization, the model avoids the intricacies and potential issues associated with traditional gradient-based training, such as vanishing or exploding gradients.

**Weaknesses:**

**Major Concerns**

1. My primary concern is the novelty of this paper. The method presented merges a prompt-decision transformer with black box prompt-tuning, but lacks a comprehensive explanation. The paper does not provide an in-depth evaluation of why trajectories were chosen as prompts, nor does it offer empirical or theoretical justifications for the suitability of the ZO-RankSGD optimization approach in this context.

2. The introduced prompt fine-tuning technique is categorized under adaptor-based fine-tuning methods. However, this paper does not offer a thorough comparison with other adaptor-based methods, such as LoRA and prefix fine-tuning, among others.

3. The experimental results seem insufficient, as only four task outcomes are reported. It would be valuable to know the performance of the compared strategies in other meta-RL tasks and to understand the pre-training environments or datasets used for PTDT.

4. Figure 3 displays the sample efficiency of the proposed method without any in-depth discussion. The results appear questionable, as all methodologies begin with high performance before fine-tuning. Notably, in the Ant-dir, PTDT's performance diminishes post-fine-tuning.

**Minor Suggestions**

1. The citation for Figure 1 is found on page 4, yet the figure itself is on page 2. It would enhance readability to position the figure closer to its mention.

2. On page 7, the author asserts that HDT is a parameter-efficient technique. This is misleading since parameter-efficient fine-tuning (PEFT) typically investigates a broad array of adaptors, inclusive of prompt-tuning.

3. The text size in Figure 3 is too small to read clearly.

**Questions:**

1. Please address the weaknesses mentioned above.
2. Page 6 line 1 "RL environments (offline and online)", what do offline and online RL environments mean?

---

### Official Review · Reviewer_pE8y · 2023-11-01

**Soundness:** 2 fair
**Presentation:** 1 poor
**Contribution:** 2 fair
**Rating:** 3
**Confidence:** 5

**Summary:**

This work proposes a Prompt-Tuning Decision Transformer (PTDT) to adapt pre-trained DT to new environments with prompting (additional inputs to model). Instead of providing K transitions as an additional context, this work utilizes a noisy version of trajectory. Specifically, given the original trajectory (prompt), the authors generate several trajectories by adding Gaussian noises and then define a ranking based on the score function (MSE loss in offline setup and Reward value in online setup). Based on the ranking graph, they approximate a gradient and add it to the original trajectory. By repeating this process, they define prompts. The authors evaluated PTDT in several locomotion and manipulation tasks.

**Strengths:**

* The authors provide a solution for adaptation based on prompting.

* PTDT achieves comparable performance with full fine-tuning models, demonstrating the promising application of this approach.

**Weaknesses:**

* Poor writing: overall, it is hard to understand the main method clearly and several components are also unclear (see Questions for more details).

* Unclear motivation: basically, the authors propose a very heuristic prompting method that defines initial inputs as a concatenation of transitions and performs an evolutional search (adding Gaussian noises to input and approximating the gradient to increase the score function). For me, it is unclear why such heuristics can be better than full-finetuning or other baselines like adaptor and soft-prompt.

**Questions:**

* At the test time, PTDT receives $x_T$ as additional inputs. During pre-training, how these additional inputs are defined? More detailed explanations on pre-training are required (i.e., how to set prompts at the pre-training stage).

* Detailed explanations about baselines and experimental setup: There are three baselines (Soft-prompt, Adaptor, full fine-tuning) but there is no clear explanation about their details. The authors need to clarify the details including # of test samples for fine-tuning, objective function, and architectures (e.g., how to model soft prompt, how to add adaptor).

* ranking functions in 6 & 7: for computing these ranking functions, how many offline trajectories and online interactions are used? There is no clear explanation about this part and my concern is on using more test data only for the proposed method.

* Figures 2 & 3: please increase the font size

* computational efficiency: the authors need to discuss the actual training time for each method.

* Related work: prompting sounds quite relevant to one-shot imitation learning [1]. It would be nice if the authors could add discussions about connection with one-shot imitation learning.

[1] Duan, Y., Andrychowicz, M., Stadie, B., Jonathan Ho, O., Schneider, J., Sutskever, I., Abbeel, P. and Zaremba, W., 2017. One-shot imitation learning. Advances in neural information processing systems, 30.

---

### Official Review · Reviewer_WG8A · 2023-11-09

**Soundness:** 3 good
**Presentation:** 3 good
**Contribution:** 2 fair
**Rating:** 5
**Confidence:** 3

**Summary:**

This paper studies the problem of prompt tuning for decision transformers. The setting is a meta-reinforcement learning setting where the objective is to use a few examples of behaviors quickly adapt a pre-trained model to a new task, which might be different from the training tasks. The authors propose a parameter-efficient method for handling this setting: preprend a ‘prompt’ to the decision-transformer which can be optimized directly. This is derived from ideas from NLP literature which do the same thing (e.g. prompt tuning, prefix tuning, etc). This paper also includes a preference-based optimization framework to tune the prompts.

**Strengths:**

- The method appears to be competitive and outperforms comparable approaches such as Prompt-DT (however prompt-dt only uses in-context learning) and more expensive finetuning approaches that propagate through the full model.
- The approach seems to be simple and easy to implement given existing work.
- The presentation is mostly good (with some exceptions listed below) and I believe the paper sufficiently discusses related work.

**Weaknesses:**

- The approach is a bit incremental. It's very similar to ideas that have previously been explored such as Prompt-DT (which is cited and discussed well) and is a natural extension of existing ideas from NLP. However, the results are good at least, which is still a plus.
- From the presentation, it’s not clear where many of the benefits come from and these were not well explored. I have questions regarding the implementation decisions listed below.

**Questions:**

- Why use a preference optimization objective when one could use any number of zero-order optimization approaches that also do not require full back-propagation? Even policy gradient could perhaps work. It would be nice to see some sort or comparison between these approaches to justify the design decision.
- How does the expert prompt initialization work for PTDT-online? Isn’t the prompt from online interactions with the simulator?
- How much more expensive is directly finetuning the prefix (e.g. Li and Liang)? This seems sufficient for many tasks in language modeling and I suspect the architecture for RL is substantially smaller.
- I’m having trouble understanding the necessity of the Gaussian noise that is added. It says this avoids ‘catastrophic deviations’ but what does this actually mean? Is this just for exploration purposes of the prompt optimization or is there another reason?
- Why is does full finetuning and soft-prompt do worse than PTDT? I feel these should be strictly better (despite being more expensive), no?


Minor:
- The figure texts are very small while the figures are very large. Increasing the text size would make this easier to read.
- ‘Preliminary’ -> Preliminaries, ‘Experiment’ -> Experiments